# Myricetin Acts as an Inhibitor of Type II NADH Dehydrogenase from *Staphylococcus aureus*

**DOI:** 10.3390/molecules29102354

**Published:** 2024-05-16

**Authors:** Jia-Le Zhou, Hao-Han Chen, Jian Xu, Ming-Yu Huang, Jun-Feng Wang, Hao-Jie Shen, Sheng-Xiang Shen, Cheng-Xian Gao, Chao-Dong Qian

**Affiliations:** College of Life Sciences, Zhejiang Chinese Medical University, Hangzhou 310053, China; xiaozhoujiayou888@163.com (J.-L.Z.); chenhaohan0915@163.com (H.-H.C.); x1697490209@163.com (J.X.); 15160486520@163.com (M.-Y.H.); 15655456852@163.com (J.-F.W.); haojietime@foxmail.com (H.-J.S.); ssx19550222949@163.com (S.-X.S.); zjtcmgcx@126.com (C.-X.G.)

**Keywords:** myricetin, *Staphylococcus aureus*, Type II NADH dehydrogenase, competitive inhibitor, anti-virulence activity

## Abstract

Background: *Staphylococcus aureus* is a common pathogenic microorganism in humans and animals. Type II NADH oxidoreductase (NDH-2) is the only NADH:quinone oxidoreductase present in this organism and represents a promising target for the development of anti-staphylococcal drugs. Recently, myricetin, a natural flavonoid from vegetables and fruits, was found to be a potential inhibitor of NDH-2 of *S. aureus*. The objective of this study was to evaluate the inhibitory properties of myricetin against NDH-2 and its impact on the growth and expression of virulence factors in *S. aureus*. Results: A screening method was established to identify effective inhibitors of NDH-2, based on heterologously expressed *S. aureus* NDH-2. Myricetin was found to be an effective inhibitor of NDH-2 with a half maximal inhibitory concentration (IC_50_) of 2 μM. In silico predictions and enzyme inhibition kinetics further characterized myricetin as a competitive inhibitor of NDH-2 with respect to the substrate menadione (MK). The minimum inhibitory concentrations (MICs) of myricetin against *S. aureus* strains ranged from 64 to 128 μg/mL. Time–kill assays showed that myricetin was a bactericidal agent against *S. aureus*. In line with being a competitive inhibitor of the NDH-2 substrate MK, the anti-staphylococcal activity of myricetin was antagonized by MK-4. In addition, myricetin was found to inhibit the gene expression of enterotoxin SeA and reduce the hemolytic activity induced by *S. aureus* culture on rabbit erythrocytes in a dose-dependent manner. Conclusions: Myricetin was newly discovered to be a competitive inhibitor of *S. aureus* NDH-2 in relation to the substrate MK. This discovery offers a fresh perspective on the anti-staphylococcal activity of myricetin.

## 1. Introduction

*Staphylococcus aureus* is a Gram-positive opportunistic pathogen that colonizes in about one-third of the general population [1]. While this pathogen typically does not infect healthy individuals, its entry into internal tissues or the bloodstream can lead to various complications, ranging from wound infections to more severe and life-threatening conditions such as septic arthritis, pneumonia, gastroenteritis, bacteremia, osteomyelitis, and meningitis [2,3,4]. Of particular concern is the escalating prevalence of antibiotic-resistant strains, notably methicillin-resistant *S. aureus* (MRSA) [5]. Infections caused by MRSA are widespread and associated with high mortality rates. Additionally, *S. aureus* ranks among the most frequent causes of bacterial foodborne diseases globally [4,6]. Consumption of food contaminated with *S. aureus* can induce symptoms of acute gastroenteritis, including diarrhea, vomiting, and fever. The pathogenicity of *S. aureus* primarily arises from its capacity to generate a variety of toxins. These toxins act synergistically, allowing *S. aureus* to evade host immune responses during infections and subsequently induce a spectrum of diseases [2,3,4,5,6]. Therefore, there is a growing interest in identifying small molecules and compounds that can disrupt the virulence mechanisms of *S. aureus*, including toxin production [4].

Type II NADH dehydrogenase (NDH-2) is a monotopic membrane flavoprotein that catalyzes the transfer of electrons from NADH to quinone via FAD [7]. This respiratory enzyme contributes indirectly to the generation of proton motive force and helps to maintain the NADH/NAD^+^ redox balance. Given its essential role in ATP synthesis or growth in certain pathogens and its absence in mammals, NDH-2 has emerged as a promising antibiotic target [7]. Different from mammals and many pathogens, *S. aureus* lacks type I NADH dehydrogenase but possesses two NDH-2 isoforms, namely, NdhC and NdhF [8]. These NDH-2s represent the sole NADH: quinone oxidoreductases in this organism. The impact on the growth of *S. aureus* in Mueller–Hinton medium was notably pronounced when NdhC was absent, while the absence of NdhF exhibited minimal effects [9]. This observation underscores the predominant role of NdhC as the primary dehydrogenase under such conditions. Importantly, inactivation of NDH-2, particularly NdhC, resulted in a significant compromise in the pathogenicity and virulence of *S. aureus* [9]. Consequently, targeting NDH-2 has been proposed as a potential approach to combat *S. aureus* infections.

Many compounds with diverse structures, such as phenothiazines [10,11,12], quinolones [13,14], quinolinyl pyrimidines [15,16], 2-mercapto-quinazolinones [17], and tricyclic spiroLactams [18], have been identified as potent antibacterial agents targeting NDH-2 in *Mycobacterium tuberculosis*. However, there is limited information available on inhibitors that target *S. aureus* NDH-2. Currently, only 2-heptyl-4-hydroxyquinoline-N-oxide (HQNO) and three phenothiazine compounds (thioridazine, trifluoperazine, and chlorpromazine) have been characterized as inhibitors of *S. aureus* NDH-2 in vitro [8,19]. Recent screening efforts of natural compounds derived from plants have unveiled myricetin, a 3, 3′, 4′, 5, 5′, 7-hexahydroxy flavone (Figure 1), as a more potent in vitro inhibitor of *S. aureus* NDH-2 compared to HQNO. This discovery prompted our investigation into the inhibitory properties of myricetin against NDH-2. Therefore, this study aims to explore the inhibitory effect and mechanism of myricetin on NDH-2, as well as its impact on the growth and virulence factor expression in *S. aureus*. 

## 2. Results

### 2.1. Protein Purification and Enzymatic Characterization

The gene encoding NADH dehydrogenase II protein (NdhC) was cloned into plasmid pET28a (+) and transformed into *E. coli* BL21(DE3). The protein was induced with isopropyl β-D-1-thiogalactopyranoside (IPTG) at 25 °C for 8 h. Consistent with the previous report [20], cells expressing NDH-2 were light greenish in color compared to the cream color of the wild-type pellet. The bacterial cells were disrupted using a pre-cooled French Press and sonicated on ice for 20 min. It is worth mentioning that the soluble protein content of NDH-2 showed a significant increase after ultrasound treatment. One explanation for this phenomenon is that the process might enhance the product release from the cell membrane, as NDH-2 is a monotopic membrane protein anchored to the lipid bilayer. Furthermore, we observed that the stability of NDH-2 was greatly improved by adding 500 mM NaCl and 5% (*v*/*v*) glycerol to the lysis and elution buffers.

The His-tagged NDH-2 was then purified using an affinity chromatography column and analyzed by SDS-PAGE (Figure 2A, Appendix A). The purified enzyme showed a molecular mass of approximately 42 kDa, in line with the predicted molecular weights of NDH-2, which was confirmed by Western blotting using an anti-His antibody (Figure 2B, Appendix A). The full-length gel and blot images are shown in Appendix A. The UV–visible spectra of the purified NDH-2 proteins showed no appearance peak near 450 nm, indicating that the enzyme carried little or no FAD, which has an absorption maxima at 450 nm in its oxidized form. Flavoproteins that are heterologously expressed commonly experience FAD loss [21]. Fortunately, the addition of exogenous FAD to the reaction mix restored the NADH dehydrogenase activity of the recombinant protein. As the purified NDH-2 contained sub-stoichiometric amounts of FAD, subsequent assays were performed with the addition of 20 μM FAD to the reaction system. 

Optimal enzyme activity also required 500 mM NaCl and 5% glycerol. NADH:quinone oxidoreductase activity was tested in the presence of NADH and water-soluble menadione (MK) as an oxidant. The results showed that the substrate NADH or MK had a dose-dependent effect on the enzyme activity of NDH-2 (Figure 2C,D). The turnover number under Vmax conditions estimated for the reaction was 15.2 ± 3.1 NADH/s, and the Km values for MK and NADH were 19.9 ± 0.7 µM and 90.8 ± 16.2 µM, respectively (Figure 2E–G). These values were close to those reported in the literature [19]. Thus, the purified NDH-2 was used to screen inhibitors of the NDH-2 enzyme.

### 2.2. Myricetin Acted as a Competitive Inhibitor of NDH-2 Substrate Menadione

We used 96-well plates to conduct the screens with a reaction mixture consisting of 2.5 μg/mL purified NDH-2 and various drugs at a concentration of 32 μg/mL. To validate the screenings, enzyme inhibition assays were performed with 3.2% DMSO (negative control) and HQNO, a well-known NDH-2 inhibitor (positive control) [22]. As shown in Figure 3A, 32 μg/mL HQNO almost completely inhibited the NADH oxidation catalyzed by NDH-2. Similar results were observed for myricetin. To determine the half maximal inhibitory concentration (IC_50_) of myricetin, different concentrations of drugs were added into the assay mixture, and the reaction was monitored via absorbance at 340 nm (Figure 3B,C). Probit analysis was performed on the data to obtain probability of response curves (Figure 3D,E). The calculated IC_50_ of myricetin was found to be 0.7 μg/mL (2 μM), which was about 4.5 times lower than that of the positive agent HQNO (2.3 μg/mL or 9 μM).

In order to understand the molecular interactions between the inhibitor and enzyme, a molecular docking study was performed. The binding poses were evaluated and ranked using a scoring function. The top-ranked binding pose of myricetin was found around the binding pocket of substrate MK (Figure 4A). The docking simulation revealed that myricetin interacted hydrophobically with several amino acid residues, including T48, F168, M209, A319, Q320, T352, K379, I382, and D383. These residues partially overlapped with the binding sites of the MK substrate [19], suggesting that myricetin acted as a competitive inhibitor of NDH-2 in relation to the substrate MK. The inhibition mechanism of the NDH-2 inhibitor was subsequently investigated by checking the residual enzymatic activity of NDH-2 at different concentrations of MK under a set of fixed myricetin concentrations (Figure 4B). As shown in Figure 4C, all double reciprocal plots had intersecting lines on the Y-axis, confirming that myricetin competitively inhibited NDH-2 for substrate MK. 

### 2.3. The Anti-Staphylococcal Activity of Myricetin Was Antagonized by MK

The sole respiratory NADH dehydrogenase found in *S. aureus* is NDH-2 [8], which facilitates electron transfer by reducing menaquinone to menaquinol. Deletion of the *ndh* gene significantly impaired staphylococcal growth when compared to the parent strain in MH medium [9]. To assess the antibacterial activity of the biochemically active NDH-2 inhibitor myricetin, we determined its ability to inhibit the growth of *S. aureus.* As shown in Table 1, myricetin showed antibacterial activity against *S. aureus* with MICs ranging from 64 to 128 μg/mL. To evaluate the bactericidal/bacteriostatic behavior of the flavonoid, time–kill assays against *S. aureus* ATCC 25923 and 43300 were performed. As shown in Figure 5A,B, no living cell was detected at 24 h for the two tested strains treated with 4 × MIC of myricetin, indicating that this flavonoid was a bactericidal agent against *S. aureus*. Subsequent research revealed that adding menaquinone-4 (MK-4) to the culture medium rescued the growth defects caused by myricetin (Figure 5C,D), which was in agreement with the above finding that myricetin acted as a competitive inhibitor of the NDH-2 substrate MK.

### 2.4. Myricetin Exhibited a Complex Impact on the Virulence Factors of S. aureus

*S. aureus* infections heavily rely on the production of a series of toxins. A recent study has demonstrated that the inactivation of NDH-2 significantly impacts the production of factors contributing to staphylococcal virulence, such as staphyloxanthin and α-hemolysin [9]. To assess the effect of myricetin on staphylococcal virulence, we first determined its impact on the gene expression of the two above-mentioned toxins using quantitative real-time PCR (qRT-PCR). Myricetin did not significantly affect the expression of *crtM* in *S. aureus* (Figure 6A). The *crtM* gene encodes dehydrosqualene synthase, which catalyzes the initial step of the biosynthetic pathway of staphyloxanthin. Conversely, myricetin marginally but significantly upregulated the expression of the gene *hla*, responsible for encoding α-hemolysin (Figure 6B). The production of α-hemolysin was further quantitatively analyzed by assessing the lysis of rabbit red blood cells, which are highly susceptible to hemolysis induced by α-hemolysin. Inconsistent with the findings from the qRT-PCR assay, myricetin inhibited the hemolytic activity of *S. aureus* at concentrations ranging from 2 to 16 μg/mL, without interfering with growth (Figure 6C,D). Such results were understandable given that myricetin has been reported to directly interfere with the hemolytic activity of α-hemolysin [23]. 

In addition, the impact of myricetin on the gene expression of staphylococcal enterotoxins (SEA and SEB) was investigated using qRT-PCR. As depicted in Figure 6E, myricetin inhibited the expression of *seA* in a dose-dependent manner within the concentration range of 2–16 μg/mL. Conversely, only minor or non-significant effects of the compounds on *seB* expression were observed (Figure 6F). 

## 3. Discussion

Myricetin is a plant-derived flavonoid compound famous for its versatile activity, and widely found in vegetables, fruits, nuts, tea, and wine [24,25]. It has been shown to have antimicrobial activity against *Mycobacterium tuberculosis* H37Rv, *Streptococcus mutans*, *Bacillus subtilis*, *E. coli*, *Pseudomonas aeruginosa*, and *Helicobacter pylori*, with a MIC of 50, 512, 250, 500, 500, and 160 μg/mL [26,27,28,29,30], respectively. Myricetin has also been reported to be capable of inhibiting biofilm formation and virulence factors produced by *S. mutans*, *S. aureus*, and *Streptococcus suis*, without interfering with growth [23,26,31,32]. Several molecular targets have been reported to be associated with the anti-infective effects of the flavonoid compound. Myricetin has been characterized as an inhibitor of various DNA polymerases, RNA polymerases, and reverse transcriptases [33,34]. It has also demonstrated significant inhibitory activity against *E. coli* DnaB helicase with an IC_50_ of 11.3 ± 1.6 μM [35]. Recently, myricetin has been identified as an inhibitor of sortase A [23,27,36], effectively attenuating the adhesion and biofilm formation of *S. mutans* and *S. aureus.* The IC_50_ of this flavonoid against SrtA from *S. mutans* and *S. aureus* has been determined to be 4.63 and 48.66 μM, respectively. Here, myricetin was identified for the first time as an effective inhibitor of NDH-2 from *S. aureus* with an IC_50_ of 2 μM. Through in silico prediction and enzyme inhibition kinetics, myricetin was further characterized as a competitive inhibitor of NDH-2 in relation to the substrate MK.

NDH-2 enzymes in *S. aureus* are crucial for the regeneration of NAD^+^, redox balance maintenance, and energy production [9]. NDH-2 inactivation impairs staphylococcal growth and reduces biofilm formation and hemolytic activity, but enhances staphyloxanthin production [9]. To investigate the potential targeting of NDH-2 in *S. aureus* by myricetin, we evaluated its effect on bacterial growth and the expression of virulence factors. As anticipated, myricetin exhibited good activity against *S. aureus*. Importantly, the anti-staphylococcal activity of myricetin was antagonized by MK-4, which is an NDH-2 substrate as an electron acceptor. The impact of myricetin on the virulence of *S. aureus* is multifaceted. At sub-inhibitory concentrations, myricetin was observed to have no effect on the expression of *crtM* and *seB*, but caused a small but statistically significant inhibition on the gene expression of enterotoxin *seA*. Although myricetin slightly stimulated the gene expression of α-hemolysin, it significantly attenuated the hemolytic activity induced by *S. aureus* culture on rabbit erythrocytes in a dose-dependent manner. This may be due to the fact that myricetin can bind directly to certain amino acids of α-hemolysin, even at concentrations well below the threshold for growth inhibition, thereby disrupting the toxin’s hemolytic activity [23]. The regulation of virulence factors in *S. aureus* is subject to a complex network that integrates cues from the host and environment into a coordinated response [37]. It is noteworthy that numerous antibiotics have been documented to exert an impact on the production of toxins by *S. aureus* when present at sub-inhibitory concentrations [38,39]. At lower concentrations, myricetin altered the virulence of *S. aureus*, despite the requirement of higher concentrations to inhibit bacterial growth. Therefore, myricetin appears to possess greater potential as a candidate for the development of an anti-virulence compound. 

In response to the growing threat of antibiotic resistance, the compounds that neutralize bacterial toxins or block the pathways regulating toxin production have received increasing attention [4,40,41,42]. Compared to traditional antibiotics, anti-toxin compounds have less selective pressure to evolve resistance because they affect bacterial virulence rather than cell viability. Although no anti-virulence approaches have yet been approved for clinical use, a growing body of evidence points to their potential benefits and efficacy, particularly in the context of infections caused by resistant bacterial strains. Future studies should prioritize in-depth mechanistic analysis of carefully selected, highly potent virulence candidates to advance their potential therapeutic applications against *S. aureus* infections.

## 4. Methods

### 4.1. Strains and Chemicals

*S. aureus* ATCC 25923, *S. aureus* ATCC 29213, *S. aureus* ATCC 43300, and *S. aureus* Newman were used in this study. *S. aureus* strains were routinely grown in Luria–Bertani (LB) media at 37 °C. Mueller–Hinton (MH) medium (Oxoid, Hants, UK) was used for drug susceptibility testing. A concentration of 50 μg/mL kanamycin was employed for plasmid selection and maintenance, as required. Myricetin (>98%, HPLC) was purchased from aladdin (Shanghai, China), while Menadione (>98%, HPLC) and NADH (>98%, HPLC) were purchased from Macklin (Shanghai, China). Myricetin was dissolved in Dimethyl sulfoxide (DMSO) and then diluted to ensure a final DMSO concentration of <3.2% (*v*/*v*).

### 4.2. Expression Plasmid Construction

For heterologous expression of NDH-2, the *ndhC* gene [WP 002463246.1] was amplified from chromosomal DNA of *S. aureus* 25923 using primers NdhC-F/NdhC-R (Table 2), and inserted into the plasmid pET28a(+) using a One-Step Seamless Cloning kit (Accurate Biology, Changsha, Hunan, China). To facilitate purification, a 6His-tag was introduced at the N-terminus of the NDH-2 gene. The recombinant plasmid was transformed into *Escherichia coli* DH5α. The transformants were spread on LB plates containing kanamycin at 37 °C, and the resulting strains were then selected and confirmed by PCR using two pairs of primers: YZ-1-F/YZ-1-R and YZ-2-F/YZ-2-R (Table 2). After confirmation by DNA sequencing, the recombinant plasmid pET28a(+)-NdhC was transformed into *E. coli* BL21(DE3).

### 4.3. Protein Preparation

The overnight cultures of *E. coli* BL21(DE3) were diluted into fresh LB media at a dilution of 1:100 and grown with 250 rpm shaking till an OD_600_ of 1 was reached. Gene expression was induced by the addition of isopropyl β-D-1-thiogalactopyranoside (IPTG) at a concentration of 1 mM to the media. The induced culture was further incubated at 25 °C for 8 h and then harvested in a centrifuge at 4 °C. All the purification procedures were carried out at 4–8 °C. Cell pellets were resuspended in lysis buffer (50 mM Tris-HCl, 500 mM NaCl, 5% glycerol, pH 8.0) and disrupted by passing three through a high-pressure homogenizer (EmulsiFlex-C5, Avestin, Ottawa, ON, Canada) at a pressure of 15,000 psi. The lysate underwent an additional 20-min ultrasound treatment (SK2510HP, Shanghai, China) in an ice bath, followed by centrifugation at 8000 rpm for 30 min to eliminate cell debris. The supernatant was added to Ni-NTA resin (Ni Bestarose FF, Jiaxing, Zhejiang, China) pre-equilibrated with lysis buffer. The resin was washed with lysis buffer plus 30–80 mM imidazole, and then the bound proteins were eluted with lysis buffer containing 150 mM imidazole. The purified protein was loaded on SDS-PAGE for the purity and confirmed by Western blotting using an anti-His antibody (HA1006, Hangzhou, China).

### 4.4. Enzyme Activity Assay

Enzyme activity assays were determined using a UV-Vis spectrophotometer (Technologies model 8453, Santa Clara, CA, USA) by monitoring absorbance at 340 nm (NADH ε = 6.22 mM^−1^ cm^−1^). The reaction mixture (200 μL) contained 0.5 μg of the purified NDH-2, 20 μM FAD, 300 μM NADH, 500 mM NaCl, 5% glycerol, and 50 mM Tris-HCl buffer, pH 7.5. The reaction was initiated by adding 30 μM menadione (MK). For determination of kinetic constants, 0 to 400 μM of NADH and 0 to 60 μM of MK were added to the reaction mixture. Kinetic parameters were calculated by nonlinear regression analysis using GraphPad Prism Software 8.2 (GraphPad Software, La Jolla, CA, USA). All experiments were performed with three replicates.

### 4.5. Molecular Docking Simulation

Molecular docking using AutoDock Vina [43] was employed to explore the molecular basis of myricetin interaction with NDH-2. The 3D structure of the small molecule was obtained from the PubChem database (https://pubchem.ncbi.nlm.nih.gov (accessed on 25 August 2022)), and the crystal structure of NDH-2 (PDB ID: 5NA1) from *S. aureus* RF144 was obtained from the PDB database (http://www.rcsb.org (accessed on 25 August 2022)). 

### 4.6. Antimicrobial Activity Assays 

Antimicrobial activity assays were performed by the minimum inhibitory concentration (MIC) method and time–kill experiments, as described previously [44]. The MIC was defined as the lowest concentration of drug with invisible bacterial growth after incubation at 37 °C for 18–20 h. DMSO (3.2%) was used as a negative control, and vancomycin was used as a positive control of bactericidal antibiotic.

### 4.7. Quantitative Polymerase Chain Reaction

The levels of expression for *gyrb*, *hlα*, *crtM*, *seA*, and *seB* were analyzed using reverse transcription (RT) followed by real-time quantitative PCR (qPCR). Briefly, a total of 2 mL of *S. aureus* grown with different concentrations of myricetin (0 to 32 μg/mL) was incubated at 37 °C with shaking at 200 rpm for 16 h. *S. aureus* grown without myricetin was treated in the same manner and used as a negative control. Subsequently, each sample was centrifuged at 10,000× *g* for 10 min to collect the bacteria. RNA was extracted from each culture using the SteadyPure RNA Extraction Kit (Accurate Biology, Changsha, Hunan, China), and its purity and concentration were determined using a SpectraMax ABS Puls (Molecular Devices, Shanghai, China). The RNA was then reverse-transcribed using the Evo M-MLV RT Mix Kit (Accurate Biology, Changsha, Hunan, China). The resulting cDNA was quantified using 2 × SYBR Green Abstart qPCR Mix from Sangon Biotech in a qTOWER3 G real-time PCR system (Analytic Jena, Jena, Germany). The primers used are listed in Table 2. The RNA transcript levels were determined using the 2^−ΔΔCT^ method. The experiment was conducted in triplicate for each reaction.

## 5. Conclusions

Myricetin was characterized as a competitive inhibitor of NDH-2 in relation to the substrate MK. In accordance with this, the antimicrobial activity of myricetin against *S. aureus* was attenuated by MK-4. The impact of myricetin on the production of *S. aureus* virulence factors is multifaceted. Most observations seem to be consistent with the finding that myricetin acts as a competitive inhibitor of NDH-2. However, further experiments are needed to clarify whether NDH-2 is the antibacterial target for myricetin. 

## Figures and Tables

**Figure 1 molecules-29-02354-f001:**
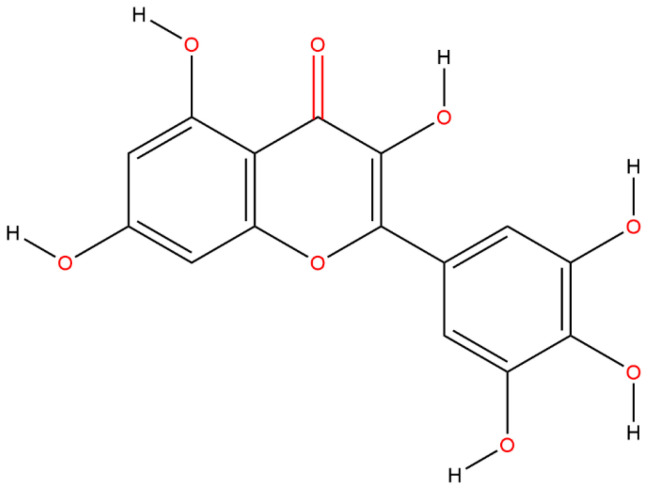
Structure of myricetin.

**Figure 2 molecules-29-02354-f002:**
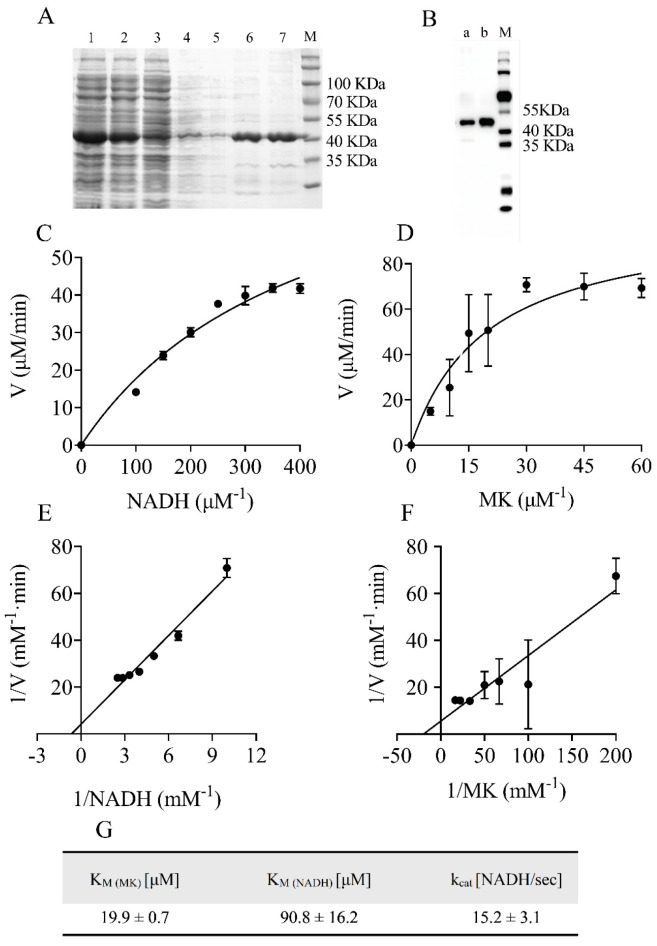
Preparation and enzymatic properties of heterologously expressed NDH−2. (**A**) SDS−PAGE of purified protein. 1: Lysate, 2: Supernatant, 3: Pellet, 4: Wash, 5: 30 mM imidazole, 6: 80 mM imidazole, 7: 150 mM imidazole. (**B**) Western blotting of NDH−2 using anti−6His−tag antibody. a: 80 mM imidazole, b: 150 mM imidazole, M: Protein Marker. (**C**,**D**) The plot of the reaction rate (V) vs. substrate concentration. (**E**,**F**) The kinetic parameters Km were calculated by Lineweaver−Burk plot of 1/V vs. 1/S. (**G**) The turnover number (kcat) and affinity (Km) for the substrates of the enzyme were determined. Data are expressed as mean ± SD of three independent experiments.

**Figure 3 molecules-29-02354-f003:**
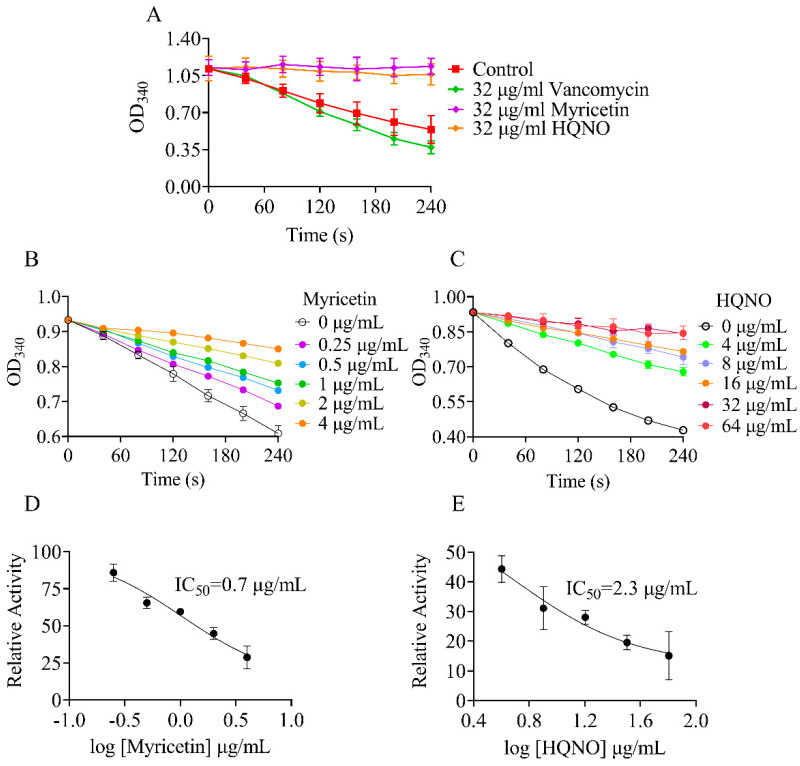
The effect of myricetin on enzyme activity of NDH−2. Concentration of NADH was set at 300 μM for all experiments. A variable slope model was fitted to determine the IC_50_ values. In each panel, NADH: quinone oxidoreduction activities in the absence of drugs were used for 100%. (**A**) The NADH UV absorbance at 340 nm was monitored after the addtion of various agents; (**B**) The NADH UV absorbance at 340 nm was monitored after the addtion of myricetin; (**C**) The NADH UV absorbance at 340 nm was monitored after the addtion of HQNO; (**D**) Myricetin inhibition curves for NDH−2 in the presence of 30 μM MK; (**E**) HQNO inhibition curves for NDH−2 in the presence of 30 μM MK.

**Figure 4 molecules-29-02354-f004:**
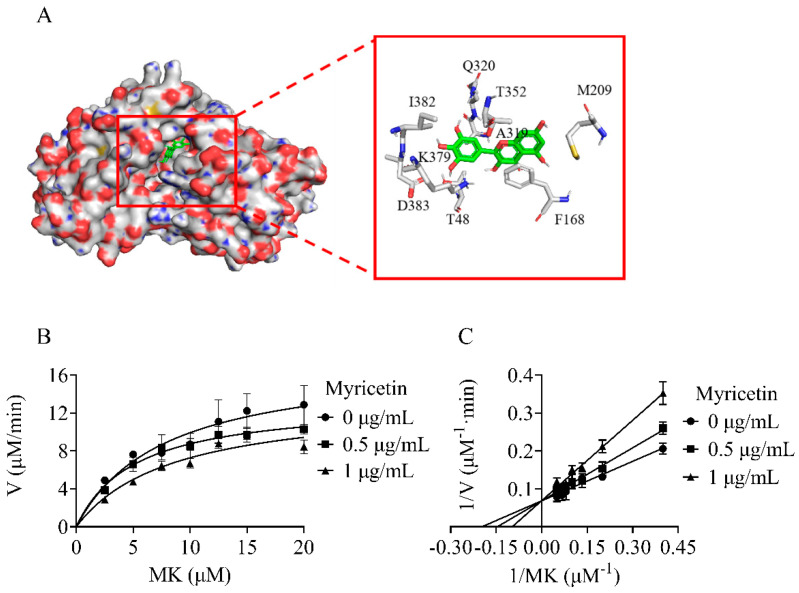
Myricetin acting as a competitive inhibitor of NDH−2 substrate MK. (**A**) Electronic cloud plot of NDH−2 and the interaction of myricetin with NDH−2 (PDB code 5AN1). (**B**) Michaelis−Menten plot of myricetin against NDH−2. (**C**) Lineweaver−Burk plot of myricetin against NDH−2.

**Figure 5 molecules-29-02354-f005:**
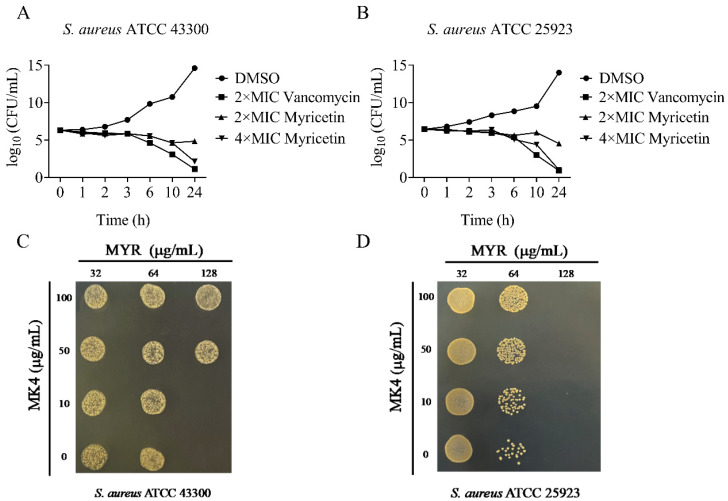
The antimicrobial activity of myricetin against *Staphylococcus aureus*. (**A**,**B**) Time–kill curves for *Staphylococcus aureus* treated with myricetin. The concentrations are as follows: ●, 3.2% DMSO; ▴, 2 × MIC of myricetin; ▾, 4 × MIC of myricetin; ■, 2 × MIC of vancomycin. (**C**,**D**) MK4 antagonizes the antibacterial activity of myricetin.

**Figure 6 molecules-29-02354-f006:**
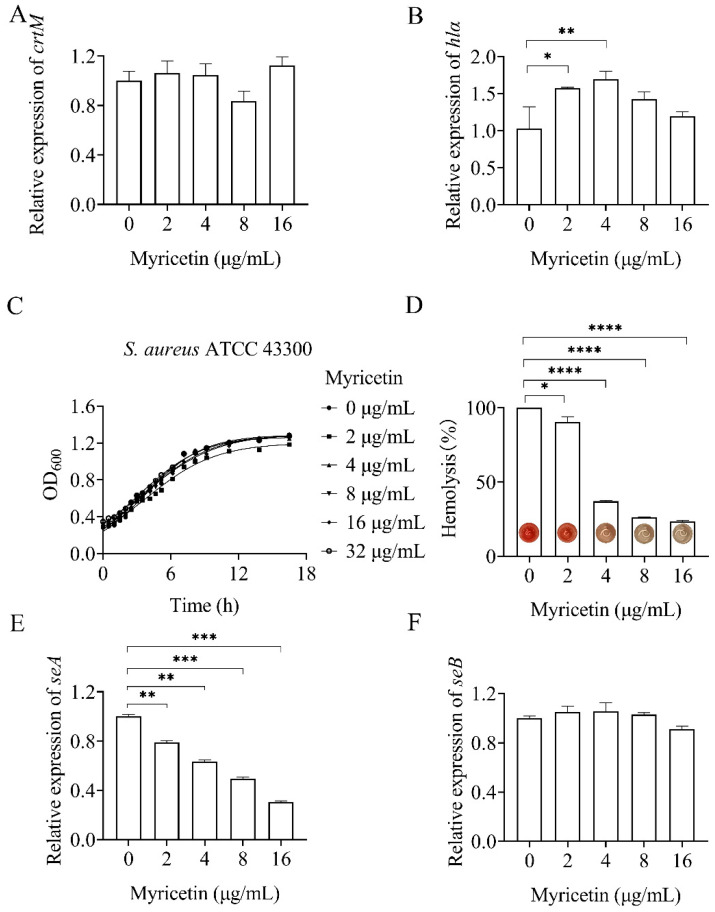
Effect of myricetin on *Staphylococcus aureus* toxins. Changes in the transcript levels of crtM (**A**), hlα (**B**), seA (**E**), and seB (**F**) in myricetin-treated S. aureus were analyzed relative to those in untreated control cultures. Relative normalized expression values were determined by RT-qPCR. (**C**) Bacterial growth was measured by optical density at 600 nm. (**D**) Hemolytic activity of myricetin-treated *Staphylococcus aureus*. Data represent means ± SD (*n* = 3). Statistically significant differences between differences in drug concentrations are shown as * *p* < 0.1, ** *p* < 0.01, *** *p* < 0.001 and **** *p* < 0.0001.

**Table 1 molecules-29-02354-t001:** MICs of myricetin against *Staphylococcus aureus* strains.

Strains	MIC (μg/mL)		
	Myricetin	HQNO	Vancomycin
*S. aureus* ATCC 43300	128	8	2
*S. aureus* ATCC 25923	128	8	1
*S. aureus* RN4220	64	4	1
*S. aureus* Newman	64	4	1

**Table 2 molecules-29-02354-t002:** Primers used in the study.

Name	Primer Sequence (5′ to 3′)
NdhC-F	CTTGCAGACGAATGTCGGCATCACAATCTGACTGAATCTTGCTTG
NdhC-R	CGGCTTAATAGCTCACGCTATGTACAACAATAAAGCCCTTCAGTG
YZ-1-F	TAGGTTGAGGCCGTTGA
YX-1-R	CTTCAATGCCAAAGGT
YZ-2-F	GCGTGGCCAAAAATAT
YZ-2-R	GTTCCTCCTTTCAGC
*hlα*-F	ACAATTTTAGAGAGCCCAACTGAT
*hlα*-R	TCCCCAATTTTGATTCACCAT
*crtm*-F	ATCCAGAACCACCCGTTTTT
*crtm*-R	GCGATGAAGGTATTGGCATT
*sea*-F	ACGATCAATTTTTACAGC
*sea*-R	TGCATGTTTTCAGAGTTAATC
*seb*-F	ATTCTATTAAGGACACTAAGTTAGGGGA
*seb*-R	ATCCCGTTTCATAAGGCGAGT
*gyrb*-F	CAAATGATCACAGCATTTGGTACAG
*gyrb*-R	CGGCATCAGTCATAATGACGAT

## Data Availability

The original contributions presented in the study are included in the article/Appendix A, further inquiries can be directed to the corresponding author/s.

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
