# Peer review of "Myricetin Acts as an Inhibitor of Type II NADH Dehydrogenase from *Staphylococcus aureus"

_molecules, 2024, doi:10.3390/molecules29102354_

Round 1

Reviewer 1 Report

Comments and Suggestions for Authors

The manuscript written by Zhou et al reported the inhibitory activity of the natural flavonoid Myricetin against S. aureus NDH-2. The results are clearly presented and should be of interest to the scientific community.

There are only few issues to fixed:

- "in vitro" should be written in Italics

- page 2: the anti-virulence strategy, as valuable approach to counteract S.aureus infections, deserves a brief discussion. Recent references on the topic should be added: Future Medicinal Chemistry, 2021, 13(6), pp. 529–531 and International Journal of Molecular Science, , 2023, 24(5), 4872

- the quality of the figures is very low and must be improved. They are blurry and unreadable. Figure 1 should also be reduced in size.

- Table 2: S. auresus should be in the abbreviated form as in the text

Author Response

The manuscript written by Zhou et al reported the inhibitory activity of the natural flavonoid Myricetin against S. aureus NDH-2. The results are clearly presented and should be of interest to the scientific community.

There are only few issues to fixed:

- "in vitro" should be written in Italics

Response: Thank you very much for pointing out this problem. We have revised the relevant issues in the manuscript.

- page 2: the anti-virulence strategy, as valuable approach to counteract S.aureus infections, deserves a brief discussion. Recent references on the topic should be added: Future Medicinal Chemistry, 2021, 13(6), pp. 529–531 and International Journal of Molecular Science, , 2023, 24(5), 4872

Response: Thank you very much for this suggestion. We have added some relevant content to the discussion section.

- the quality of the figures is very low and must be improved. They are blurry and unreadable. Figure 1 should also be reduced in size.

Response: Thank you very much for this concern. We have made changes accordingly in the revised manuscript.

- Table 2: S. auresus should be in the abbreviated form as in the text.

Response: Thank you very much for this concern. We have made changes accordingly in the revised manuscript.

Reviewer 2 Report

Comments and Suggestions for Authors

Authors report that myricetin, one of the flavonoids, was shown to be an inhibitor of type II NADH dehydrogenase (NDH-2) from Staphylococcus aureus.
S. aureus is an opportunistic pathogenic microorganism in humans, but on some occasions, it could cause severe symptoms. Additionally, the issue of antibiotic-resistant bacteria has been a rising concern.
Type II NADH dehydrogenase (NDH-2) plays a central role in the respiratory metabolism of bacteria. The lack of NDH-2 in mammalian mitochondria and its essentiality in bacterial pathogens suggest that the enzyme has been considered to be drug target to challenge microbial pathogens.
In this paper, the results showed that the efficacy of myricetin as the inhibitor was much weaker than current drugs, however, the experiments were designed and carried out correctly, and the results are scientifically sound. Therefore, this paper presents important information on the drug design for NDH-2. However, the authors should respond to some comments below before the publication.

L66: How NDH-2s share homology between S. aureus and M. tuberculosis?
L70: As the authors say “Recent screening efforts”, is this author’s work? If so, they should describe this more, if not, they should cite references.
L105: glycerine -> glycerol
L110: imidazone -> imidazole
L117: Is this reaction condition for what assay? If for the competition, the protein concentration is different from described in line 197. If for the kinetics, it is described in line 119.

Is myricetin water-soluble or not? It is better described in somewhere materials and methods.

NDH-2 is a membrane protein, so it is assumed to use any detergent related to solubilizing the protein. Did authors consider 2 M NaCl as a substitute?

Line 157: Bacteria were -> The protein was
Line 184: According to the kinetic parameters, the Km value of MK was smaller than that of NADH. Others in references reported smaller values for NADH than those for quinones. Any discussions on this point?

In the qPCR experiment, authors used 2 to 16 ug/mL myricetin, however, MIC is much higher in Table 2. Why did the authors use these lower concentrations for the qPCR experiment?

Author Response

Authors report that myricetin, one of the flavonoids, was shown to be an inhibitor of type II NADH dehydrogenase (NDH-2) from Staphylococcus aureus.

  1. aureus is an opportunistic pathogenic microorganism in humans, but on some occasions, it could cause severe symptoms. Additionally, the issue of antibiotic-resistant bacteria has been a rising concern.

Type II NADH dehydrogenase (NDH-2) plays a central role in the respiratory metabolism of bacteria. The lack of NDH-2 in mammalian mitochondria and its essentiality in bacterial pathogens suggest that the enzyme has been considered to be drug target to challenge microbial pathogens.

In this paper, the results showed that the efficacy of myricetin as the inhibitor was much weaker than current drugs, however, the experiments were designed and carried out correctly, and the results are scientifically sound. Therefore, this paper presents important information on the drug design for NDH-2. However, the authors should respond to some comments below before the publication.

L66: How NDH-2s share homology between S. aureus and M. tuberculosis?

Response: Thank you very much for this concern. NDH-2s from different microorganism have a very low sequence identity among each other. The amino acid sequence of S. aureus NDH-2 shares < 30% identity with that of M. tuberculosis, suggesting a plausible mechanism underlying drug selectivity.

L70: As the authors say “Recent screening efforts”, is this author’s work? If so, they should describe this more, if not, they should cite references.

Response: Thank you very much for this concern.Recent screening efforts is our work. Details of the screening work will be published in subsequent manuscript.

L105: glycerine -> glycerol

Response: Thank you very much for pointing out this problem. We have made changes accordingly in the revised manuscript.

L110: imidazone -> imidazole

Response: Thank you very much for pointing out this problem. We have made changes accordingly in the revised manuscript.

L117: Is this reaction condition for what assay? If for the competition, the protein concentration is different from described in line 197. If for the kinetics, it is described in line 119.

Response: Thank you very much for this concern. This is the basic enzyme activity assay method. The enzyme concentration is not changed in either the NdhC dehydrogenase activity assay or the kinetic assay; the NdhC dehydrogenase activity assay only changes the substrate concentration, and the enzyme kinetic assay changes the substrate concentration and the concentration of the drug being tested. The reaction mixture (200 μL) containing 0.5 μg of purified NDH-2' in line 119 is the same as "with a reaction mixture consisting of 2.5 μg of NDH-2" in line 201.

Is myricetin water-soluble or not? It is better described in somewhere materials and methods.

Response: Thank you very much for this concern. We have made changes accordingly in the revised manuscript.

NDH-2 is a membrane protein, so it is assumed to use any detergent related to solubilizing the protein. Did authors consider 2 M NaCl as a substitute?

Response: Thank you very much for this concern. We did not try 2M NaCl as a solubiliser, since high concentrations of sodium chloride can affect the activity of NDH-2. We found that ultrasound can effectively solubilize NDH-2 from the membrane.

Line 157: Bacteria were -> The protein was

Response: Thank you very much for pointing out this problem. We have made changes accordingly in the revised manuscript.

Line 184: According to the kinetic parameters, the Km value of MK was smaller than that of NADH. Others in references reported smaller values for NADH than those for quinones. Any discussions on this point?

Response: Thank you very much for this concern. We are of the opinion that this is probably due to the different reaction systems, especially the use of different brands of NADH reagent.

In the qPCR experiment, authors used 2 to 16 ug/mL myricetin, however, MIC is much higher in Table 2. Why did the authors use these lower concentrations for the qPCR experiment?

Response: Thank you very much for this concern. In order to avoid the influence of bacterial growth on the determination of toxins, the determination of toxins was performed at sub MIC concentrations.